# Deep Learning Assessment for Mining Important Medical Image Features of Various Modalities

**DOI:** 10.3390/diagnostics12102333

**Published:** 2022-09-27

**Authors:** Ioannis D. Apostolopoulos, Nikolaos D. Papathanasiou, Nikolaos I. Papandrianos, Elpiniki I. Papageorgiou, George S. Panayiotakis

**Affiliations:** 1Department of Medical Physics, School of Medicine, University of Patras, 26504 Rio, Greece; 2Laboratory of Nuclear Medicine, University Hospital of Patras, 26504 Rio, Greece; 3Department of Energy Systems, Gaiopolis Campus, University of Thessaly, 41500 Larisa, Greece

**Keywords:** deep learning, feature extraction, medical imaging, biomarkers

## Abstract

Deep learning (DL) is a well-established pipeline for feature extraction in medical and nonmedical imaging tasks, such as object detection, segmentation, and classification. However, DL faces the issue of explainability, which prohibits reliable utilisation in everyday clinical practice. This study evaluates DL methods for their efficiency in revealing and suggesting potential image biomarkers. Eleven biomedical image datasets of various modalities are utilised, including SPECT, CT, photographs, microscopy, and X-ray. Seven state-of-the-art CNNs are employed and tuned to perform image classification in tasks. The main conclusion of the research is that DL reveals potential biomarkers in several cases, especially when the models are trained from scratch in domains where low-level features such as shapes and edges are not enough to make decisions. Furthermore, in some cases, device acquisition variations slightly affect the performance of DL models.

## 1. Introduction

There is extensive discussion about the quality and usefulness of the image features extracted by deep learning (DL) methods, especially in medical imaging. DL holds first place in terms of the number of features mined and quantified throughout the processing layers of a deep network; however, few studies have performed a deep analysis of the clinical, prognostic, and diagnostic value of these features. This is due to several facts, such as the massive number of features to be analysed, the general behaviour of deep networks as black boxes, and the lack of statistical and mathematical tools to distinguish significant attributes from sets containing thousands or millions of features [1].

The application of DL in processing medical images, especially in developing predictive and diagnostic models, is hindered by the fact that it is neither explainable nor transparent. Moreover, the model’s supervision of the decision-making process is unknown to engineers, let alone to medical staff. On the other hand, the discovery of vital image biomarkers has recently catalysed the emergence of many techniques based on machine learning (ML) and others that export features in a nonautomatic way [2,3]. 

Despite the significant results, the latter procedure (i.e., manual feature extraction) bears limitations. Manual feature extraction or supervised automatic predefined feature extraction works assuming no other feature exists except those under investigation. For example, research focused on revealing the prognostic or predictive value of a predefined feature or a well-known biomarker, regardless of the results, overlooks the possible presence of other features, which may be more essential than the one studied. This is time-consuming and may overlook potential correlations between discovered and undiscovered features or biomarkers. Frequently, the candidate biomarker is not a new feature discovered in the image but a prominent feature, the extraction of which is possible with conventional tools. The above issues are partially valid in radiomics [4] analysis. 

### 1.1. Related Work

Most related research in the field is focused on predicting clinical outcomes by analysing predefined potential biomarkers extracted from medical and clinical image data [5,6,7,8,9,10]. In addition, there are studies aiming to discover new biomarkers from gene expression data or other clinical features. For example, Akbari et al. [11] utilised DL features and radiomic features (subtle yet spatially complex quantitative imaging phenomic (QIP) features) from Magnetic resonance imaging scans to achieve a noninvasive classification of true progression (TP) versus non-TP and TP versus pseudo-progression (PsP). For the in-depth features, the authors utilised a pretrained network. Courtiol et al. [12] designed a network to predict the survival times of patients with malignant mesothelioma (MM) based on histological criteria. They utilised histopathological images from two publicly available image data repositories. Their methodology yielded better accuracy in predicting patient survival compared to pathology practices. In addition, the proposed network suggested regions of interest contributing to the outcome. Those regions refer to the stroma and contain histological features associated with inflammation, cellular diversity, and vacuolisation.

Discovering novel biomarkers from medical images is under-represented in medical research. Zhuang et al. [13] proposed a novel methodology for classifying autism spectrum disorder (ASD) and identifying biomarkers for ASD, utilising the connectivity matrix calculated from fMRI as the input. They employed invertible networks and determined the decision boundary. The difference between the data point and projection onto the decision boundary was utilised, on the basis of which potential biomarkers were defined. Regression was used for the validation of the suggested biomarkers. The experiments demonstrated that the proposed network is effective and interpretable, thus allowing for new biomarker discoveries.

Lei et al. [14] proposed a survival analysis system utilising DL features combined with feature selection methods and survival analysis strategies, such as the Cox proportional hazards model. Their experiments were performed utilising open lung cancer datasets. The DL framework suggested specific biomarkers during their experiments, which were analysed. Furthermore, the DL methodology suggested regions where discriminating features were discovered.

Waldstein et al. [15] analysed approximately 50,000 retinal optical coherence tomography (OCT) volume scans of roughly 1000 patients with age-related macular degeneration employing DL to generate 20 local and global candidate biomarkers. Those candidate features correlated with traditionally used biomarkers in clinical practice. In addition, they discovered hitherto unknown biomarkers. Their research demonstrated that the discovered image features are closely connected with the traditional biomarkers with the specific DL approach. A correlation analysis was performed to achieve the above.

The studies mentioned above indicate that DL methods could be useful in revealing important image features that correlate well with existing biomarkers or suggest new ones. However, the results of the conducted studies are not easily validated due to dataset size limitations. This happens because the overall importance of the selected potential biomarker cannot be verified outside of the data sample utilised for a particular study. However, a correlation between a certain disease or therapy response and a predefined feature or characteristic may exist in a specific data selection. Therefore, thousands of image data must be analysed to confirm the discovery of a new biomarker, let alone the fact that these data should be representative of the actual population.

Moreover, the inherent ability of DL to reveal such biomarkers is not formally validated. Especially for clinical image analysis, the automatic procedures must be straightforward, transparent, explainable, and reproducible [16]. Every scientific conclusion derived from an artificial intelligence (AI) method, which may affect and redefine the medical pipeline in many domains, must rely on established techniques, analytical experiments, and clinical evaluation. In medical imaging, the definition of quantifiable and reproducible significant biomarkers evolves into a particular field of research. It utilises the full range of tools developed, such as ML, DL, and radiomics. 

### 1.2. The Aim of the Study

The present study intends to assess DL methods from a biomedical engineering perspective rather than a statistical and mathematical evaluation based solely on predefined accuracy metrics. The study evaluates the potential importance of medical imaging features extracted by several DL models by examining the following:
a.The effectiveness of transfer learning based on the type of the extracted features;b.The models’ robustness to acquisition device variation;c.The validity of the extracted feature maps across the layers of the best-performing DL models;d.The post hoc interpretation of Grad-CAM visualisations presenting the suggested areas of interest.

As discussed in the remainder or the paper, these four methodological steps can offer important information about the extracted imaging features’ type and importance, regardless of the classification metrics achieved by each DL model.

The experiments of the study are performed using several medical imaging datasets, including solitary pulmonary nodule (SPN) computed tomography (CT) scans, COVID-19 CT and X-ray scans, coronary artery disease (CAD) SPECT scans, skin cancer photographs, histopathological images of lung, colon, lymphoma, and prostate cancer, and cervical cancer images. In addition, seven state-of-the-art DL models are employed and compared with a modified version of VGG19 [17] proposed by the author team in recent studies [18,19,20,21,22] that demonstrated promising results.

The contributions of this study can be summarised as follows:(i).Deep learning CNNs are benchmarked to inspect their reasoning based on the extracted features. This analysis yields strong evidence that features of medical interest are extracted when training the proposed networks from scratch.(ii).The effectiveness of the Grad-CAM algorithm in improving the explainability of deep learning models is evaluated by analysing the suggested areas of interest.(iii).An innovative modification of the VGG19 network is extensively evaluated on various image classification tasks.(iv).It is concluded that FF-VGG19 and Xception networks are the most suitable CNNs for classifying medical images from various sources, such as CT, SPECT, and X-ray.

## 2. Materials and Methods

### 2.1. DL Feature Extraction Methods

The types of features mined by a DL model depend on network structure and the combination of the consisted parameters. For example, different features are extracted by a convolution filter of 3 × 3 pixel size compared to a convolution filter of 5 × 5 pixel size. Thus, an almost infinite number of features can be extracted. It is possible that a large variety of different features contribute equally to the desired results and could theoretically be excluded by employing a feature selection method. The model learns the significance of each feature throughout the loss backpropagation method, which updates the weights assigned to each filter. In this way, features irrelevant to the desired outcome suffer weight penalties. 

While many researchers utilise handcrafted CNNs to perform medical image classification tasks, it is accepted that state-of-the-art networks are preferable since they are already validated in similar tasks. Well-established networks commonly yield better results [22,23,24,25,26]. Moreover, the manually designed networks are mostly randomly picked and adjusted following various experiments. With no doubt about their effectiveness, it is rare for the proposed networks to be dedicated to a specific project and for their parameters to be documented on the basis of a specific goal. Networks that follow this process go beyond the narrow boundaries of the domain for which they were designed and are applied to other domains [27].

In DL, employing networks trained on one image task (e.g., classification) to perform the same task on a set of different but related images is called transfer learning. This is the case in most studies employing CNNs. Transfer learning refers to the process wherein knowledge is transferred from one domain to another.

#### 2.1.1. Feature Extraction via Transfer Learning or Learning with Off-the-Shelf Features

This method is often called feature extraction via transfer learning. It involves employing a pretrained network, retaining its architecture and learned filters, and training only some extra dense layers with the domain images under analysis. As CNN maintains knowledge from the first domain, this knowledge transfer limits the learning capacity of CNN. The network attempts to reveal, extract, and rate new features from the target domain task, but it will extract information according to its initial mastery. Usually, the extracted features are validated and classified by an effective classifier, such as a neural network or a support vector machine. In various examples, the retained knowledge (i.e., weights) is adequate to estimate and correctly predict the class of an image of another (but related) domain. The images of the source and the target images share some common decisive characteristics. 

#### 2.1.2. Training from Scratch

Conversely, transfer learning with training from scratch borrows only the architecture of the CNN, not its obtained knowledge. The complete network is trained on images of the domain of interest, adjusting the network to expertise on the specific task and the filters according to the desired outcome. Training from scratch is avoided when we face the data scarcity issue, or when the CNN is so deep that it introduces millions of trainable parameters, which are impossible to learn. Training from scratch may also be avoided when the source and target images belong to the same domain and share many characteristics. In such cases, feature extraction via transfer learning may be preferable. Lastly, this learning method severely increases the computational cost, leading to long-term training on a scale of hours, days, or even months [28].

#### 2.1.3. Fine-Tuning

A hybrid learning method, which consists of the abovementioned strategies, is called fine-tuning. This method involves adjusting the trainability of specific filters of the CNN while retaining its original architecture. In this way, useful knowledge from the initial training is transferred to the new task. Some filters are randomly initiated and allowed to be adjusted during the backpropagation procedure of the training and validation process.

### 2.2. Datasets

The present study utilises 11 medical imaging datasets of various modalities and domains. Table 1 summarises the characteristics of each dataset.

#### Image Pre-Processing and Data Augmentation

Data augmentation and image processing methods are applied to normalise the images, reduce dimensionality, and prevent overfitting. 

Data augmentation methods intend to expand the training data and help the networks ignore spatial variations and irrelevant image characteristics. For example, the random rotations aim to present different representations of the same finding and instruct the network to go beyond the specific location of the finding, i.e., to detect the important finding and focus on it without considering its exact position. Random Gaussian noise helps the networks generalise because they are forced to depend their decisions on holistic features rather than specific ones. Data augmentation in medical imaging must be applied cautiously. Heavy image distortions that do not correspond to realistic representations can be avoided because they can confuse the models.

Moreover, it is often the relevant location of a finding that contributes to the final result. For example, the location of a lung nodule (e.g., near the pleura) is crucial to the diagnosis. Applying augmentations that disrupt this image characteristic may result in a faulty diagnosis.

Data augmentation is not applied before training but simultaneously during each training fold over a 10-fold cross-validation procedure, ensuring that the generated images are not included in the test sets. For each dataset, the image pre-processing parameters are described as follows:For the PET/CT and LIDC-IDRI datasets, the extracted nodule images are rescaled to 80 × 80 pixels, close to the minimum acceptable size for the CNNs. Image augmentations include 10° rotations, horizontal and vertical shifts, and random Gaussian noise additions with a mean of zero and a sigma of 0.01^0.5^.All images are placed and fit to a 266 × 200 black background for the COV-X dataset. Slight augmentations (rotation by 20°) are applied during the training. For the COV-CT dataset, augmentation refers to width and height shifts (at a maximum pixel range of 10%) and the addition of Gaussian noise with a mean of zero and a sigma of 0.01^0.5^.For the Cells and Cells2 dataset, the images are rescaled to 100 × 100 pixels, and no augmentation methods are applied.As far as the MPIP dataset is concerned, the images are rescaled to 200 × 200 pixels, and slight rotations (maximum 10°) are applied to each polar map before concatenating them into a single four-polar-map image.The images of the Leukaemia dataset are rescaled to 80 × 80 pixels, and no data augmentation is applied during training.The images of NV-LES, MNIST, and SKIN datasets are rescaled to 200 × 150 pixels, and no data augmentation is applied.

### 2.3. DL Networks

The study employs seven state-of-the-art networks that stood out in the ImageNet challenge [39]. In addition, an innovative modification of VGG19 introduced by the authors in a recent study [19] is also employed. Table 2 presents an overview of the employed CNNs. Figure 1 presents the structure of the FF-VGG19 network.

### 2.4. Research Methodology

An overview of the methodology is presented in Figure 2.

#### 2.4.1. Training from Scratch and Feature Extraction via Transfer Learning

Each network is trained using the aforementioned transfer learning methods in this experiment. The main aim of this experiment is to test the hypothesis that training from scratch is superior to feature extraction via transfer learning. Secondly, the best-performing networks are compared to define a superior network suitable for all the datasets. Table 3 summarises the involved sub-experiments. 

The reader can notice that the training dataset is the same as the test dataset during this experiment, and this was achieved following a 10-fold cross-validation procedure. 

#### 2.4.2. Investigating the Robustness of DL Methods to Image Acquisition Device Variation

Results of the previous experiment revealed that training from scratch yields better results and may contribute to potential biomarker detection from the input images. To further investigate this, a second experiment was conducted. The test datasets differed from the training datasets to investigate the robustness to variation in image acquisition devices. For example, to investigate the importance of the features extracted by the training of the CNNs on the LIDC-IDRI dataset, the CNN was asked to distinguish benign and malignant nodule images from the PET/CT dataset. This procedure was applied to three major medical classification tasks:Distinguish between benign and malignant SPNs;Distinguish between benign and cancerous cells;Distinguish between lymphocyte and monocyte blood cells.

The source and the target domain classification tasks are the same. However, since there is a variation in the data distributions of images between datasets of the same kind (e.g., SPN datasets), a sub-optimal feature extractor could be misled if the extracted features are dataset-specific. In this scenario, employing this classifier to classify images from an alternative dataset (of the same domain and task) would not be effective. On the other hand, if the trained classifier ignores the irrelevant artefacts and characteristics of the hidden dataset and achieves a high classification score, that would be a prime indicator that the classification is based on realistic and precise features (potential image biomarkers). The experiments of the second phase are explained in Table 4. 

#### 2.4.3. Feature and Output Visualisation Methodologies

The previous experiments included performance assessment in terms of quantifiable metrics. In the present experiment, the output feature maps and the Grad-CAM algorithm visualisation results were used as the criteria for rating the performance of DL models. Those two methods are explained below.

a.Grad-CAM algorithm [46]

The Grad-CAM algorithm intends to identify the areas of the input image having a critical effect on the classification decision of the classifier placed at the top of the CNN. Hence, its functionalities are fully exploited in object detection tasks, where a specific image area contains the desired object. 

b.Extracted Feature Map visualisation method

This methodology tracks the output feature maps of every CNN layer and reconstructs them into an image. In this way, the user can inspect what features are learned during the process and decide their importance. 

Visualising the areas of interest or the extracted feature maps does not solve the explainability issue. However, it is useful for reaching some conclusions. More precisely, the visualisation of the feature maps can provide relevant information as to how the DL models handle the feature extraction process, how the extracted features are produced during the convolution stages, and whether those features represent expected patterns also visible by the human eye. 

## 3. Results

In this section, the results of the study are analytically presented. Section 3.1 presents the results of transfer learning with “off-the-shelf” features. Next, in Section 3.2, the training from scratch results are discussed. In Section 3.3, the robustness of device variation is presented. In Section 3.4 and Section 3.5, the evaluations of the extracted features maps and the Grad-CAM maps are presented, respectively.

### 3.1. Transfer Learning with “Off-the-Shelf” Features

The performance of the various CNNs on the 11 medical image datasets is presented in Table 5. Each experiment was conducted three times to inspect the reproducibility of the results. A variation of ±2% was observed for each training–testing run. Hence, we considered the classification accuracies equivalent when their difference was below 2%.

The suboptimal performance observed in Table 5 underlines that the transfer learning strategy utilising off-the-shelf features may not be effective for discovering and revealing significant, deep, and high-level medical-specific features from the inputs. However, slightly better performance was expected and verified in tasks where low-level features, such as the shape of the depicted SPN, are essential predictors of the desired outcome. For example, it was observed that, for the LIDC-IDRI dataset, VGG19 achieved 80.97% classification accuracy, and its predictions were probably based on the shapes and sizes of the SPNs. Low-level features were adequate to successfully characterise 80.97% of the SPN images incorporated into LIDC-IDRI. However, this did not apply to the PET/CT dataset containing SPN images. Good classification accuracies were observed in Cells, NV-LES, and MNIST datasets, achieved by either FF-VGG19 or VGG19. This may reveal that the architecture of those networks aids in extracting deep and useful features, even if the initial training was performed using nonmedical images. This assumption could be validated in future research. In Table 6, more evaluation metrics are given for the best-performing network in each dataset. 

FF-VGG19 obtained the best accuracy in most of the experiments, specifically, 72.09% on PET dataset, 84.86% on LIDC dataset, 84.96% on COV_X dataset, 79.35% on COV_CT dataset, 90.74% on Cells dataset, 61.80% on Cells2 dataset, 77.92% on Leukaemia dataset, 63.24% on MPIP dataset, 82.36% on NVLES dataset, 77.74% on SKIN dataset, and 75.55 on MNIST dataset. The classification accuracies were also accompanied by acceptable AUC and F1-scores, as seen in Table 6. 

### 3.2. Feature Extraction via Training from Scratch

The results underline the effectiveness of training from scratch. Specifically, almost every CNN improved its accuracy in every experimental dataset, as demonstrated in Table 7.

FF-VGG19 and Xception stood out in this experiment, achieving high classification accuracies in every dataset. Training the CNNs from scratch yielded better results in most cases. Xception achieved the best accuracy on PET/CT (69.77%), and FF-VGG19 achieved the best accuracy on LIDC-IDRI (82.52%). FF-VGG19 attained the best result on the COV_X dataset, yielding 84.64% accuracy. It also achieved the best scores on COV_CT and Cells datasets, yielding 78.57% and 91.61% accuracy scores. Xception exhibited better results than any other CNN on Cells2 and Leukaemia datasets (59.66% and 78.85% accuracy, respectively). FF-VGG19 achieved optimal results on the MPIP and NVLES datasets (66.08% and 83.66% accuracy). Lastly, the best accuracy on the MNIST dataset was obtained by FF-VGG19 (77.93%).

It can be observed that training from scratch was beneficial for most networks. Furthermore, it was revealed that training from scratch allowed the models to adjust their weights to specific tasks and not base their predictions on pre-learned features.

While VGG19 appeared to yield better results when training with off-the-shelf features, the other networks showed significant improvements. For example, Xception improved its accuracy in classifying the Cells dataset from ~34% to ~91%. A visualisation of those comparisons is illustrated in Figure 3.

As illustrated in Figure 3, it can be assumed that Xception, ResNet, InceptionV3, and DenseNet121 were best for training from scratch. his is not irrelevant because those networks consist of dozens of layers. In contrast with those networks, VGG19 yielded better results when utilising a transfer learning approach. FF-VGG19 was more stable to learning strategy alterations, maintaining its performance at a top level.

FF-VGG19 was the optimal CNN for achieving maximum accuracy for most datasets, while training from scratch and transfer learning had no important impact on its performance. FF-VGG19 yielded good AUC scores for every dataset, as shown in Table 8. In Figure 4, an overview of the CNNs’ performance for both learning strategies is illustrated.

Some concluding remarks are reported as follows:CNNs failed to perform optimally in small datasets. This behaviour was expected, as CNNs require large-scale training sets to learn. The features extracted by every CNN from Cells2, MPIP, and PET/CT dataset were expected to be low-level image features, such as edges and shapes. In these cases, the features did not constitute novel biomarkers;Training from scratch or transfer learning is a suitable learning strategy in cases wherein a feature of vital medical importance can be the shape, the greyness level, or the size of a pathological finding. Preferable CNNs for small datasets include VGG19 and MobileNet;FF-VGG19 was a robust CNN for most of the medical image datasets. Therefore, it could serve as a general model for similar medical image classification tasks, either without modifying its components and parameters or performing a slight modification to improve the accuracy further.

### 3.3. Robustness to Medical Image Acquisition Device Variation

The best-performing networks (FF-VGG19 and Xception) were utilised in this experiment. Under this setup, the two CNNs were trained using a dataset of one domain (e.g., SPN malignancy identification) but were evaluated using an unseen dataset of the same domain. The classification accuracies obtained are illustrated in Table 9, while the corresponding sensitivities, specificities, AUC scores, and F1-scores are given in Table 10 and Table 11.

Both networks successfully classified the PET/CT dataset when trained using samples from the LIDC-IDRI dataset. However, the PET/CT dataset was more effectively classified when the networks used LIDC-IDRI as their training set. Specifically, training and testing FF-VGG19 on the PET/CT dataset yielded 69.77% accuracy, while training on LIDC-IDRI and predicting PET/CT yielded 74.41% accuracy. This observation shows that both networks learned to extract important features that may define an SPN in general and performed actual reasoning.

The same phenomenon was observed when the target dataset was MNIST. For example, training FF-VGG19 on MNIST and testing on the same dataset returned an accuracy of 77.96%. At the same time, training on the concatenation of SKIN and NVLES datasets yielded a prediction accuracy of 84.01% on the unseen testing dataset (MNIST). Therefore, it is fair to assume that SKIN and NVLES datasets were effective training samples which contained enough images to help FF-VGG19 acquire and learn the right underlying features.

A decrease of ~10% in accuracy was observed for the remaining experimental cases. While this performance raises concerns about the validity of the learned features, other issues may also have affected the results. For example, the soft data augmentation may have oversimplified the training sets and led the network to learn too specific information, making them unable to generalise and handle new test sets, even in the same domain. This issue needs further investigation.

### 3.4. Feature Map Visualisation

The activation maps from specific layers of the FF-VGG19 CNN were visualised to inspect the learned weight filters. For every dataset, a random image was selected and classified from the trained FF-VGG19. The training was performed on various datasets, as explained in Table 12. Figure 5 and Figure 6 illustrate the feature maps.

Figure 5 and Figure 6 include the activation maps of randomly selected images from the datasets that visualise the outputs of specific convolution layers from the FF-VGG19 network. Five sets of activation maps, including nine random feature maps each, are reported for each sample original image (labelled a, b, c, etc.). Those groups are numbered from one to five.

The outputs of the early layers, i.e., the layers which performed their analysis first (numbers two to four), were less abstract and detected basic edges and patterns of the image. The black colours depict areas with less information, and the brighter parts highlight important characteristics. Deep convolution layers seek to extract more abstract and underlying features to discover new information related to the task. Those feature maps are easily understood because they seem meaningless to the human eye and brain (maps numbered five and six). However, the investigation of this research was directed toward those feature maps, which may have hidden potential biomarkers.

The feature maps extracted from the skin cancer dataset (MNIST, SKIN, NVLES) tended to characterise the skin texture as a special feature in the image, as seen in Figure 5. Although the hardness of the skin, its roughness, and the presence of hair do not constitute predictors for the severity of skin cancer, the early feature maps detected and analysed their patterns captured in the image. Of course, the classifier’s responsibility is to properly adjust the weights to reject those features for the specific task. This ability is not irrelevant to the available data, the network’s learning capacity, and the parameter tuning. However, late convolutional layers tend to ignore such features in an attempt to go deeper. 

In Figure 5(a5), the surroundings of the SPN were separately captured. This confirms the assumption that CNNs captured both point and spatial information while performing an automatic segmentation of potential areas of interest. Furthermore, from the feature maps of the SPNs, especially in a2, a3, and a4, one can notice that the outline of the SPN, which was fainter than their main part (the centre), was also captured separately. Following this visualisation, it was assumed that the CNN attempted to separate the detected shapes into complementary shapes.

In Figure 5(b6), it can be noticed that the content behind the imaged bones, i.e., the histological information, acquired its representation in the thousands of the extracted features. The feature maps b2, b3, b4, and b5 did not capture any important information because they focused on geometries and shapes concerning the skeleton, not what was behind, around, or inside. The same phenomenon was confirmed in the case of the CT images (Figure 5c).

In cases where early detected features related to shapes or edges were less relevant for the particular classification task (e.g., myocardial perfusion imaging polar maps—MPIP dataset), the feature maps were not informative (Figure 6(a2–a4)). As seen in a5 and a6, the convolution process captured specific areas inside the polar maps. Those areas were indeed decisive for the diagnostic test outcome, despite the test’s suboptimal sensitivity and precision. Areas in green were assumed to be indicators (although weak) of the risk of coronary artery disease. However, in many cases, the test yielded false positive results, making things more complex for the diagnosis.

In Figure 6(b2), the reader can notice that the outline of the cancerous cell structure was captured, and more vital information about its texture was captured in later feature maps (b5, b6). Again, the skin texture was separately captured in b3 and b4.

The feature maps’ visualisation method is helpful in determining whether the deep convolutional process is learning important features or is at least seeking patterns in the positive direction. From the illustrated feature maps, is the following could be confirmed in most cases:Both relevant and irrelevant features were captured;Most important and decisive features were discovered in deeper convolutions;At least half of the extracted features were not of medical importance, while the significance of the remainder is yet to be investigated (excluding low-level features, which are known to be significant in specific examples).

Since it is implausible to inspect every produced feature map, especially in deep networks, it is not easy to confirm that all potential biomarkers were extracted from FF-VGG19. However, looking beyond a specific network and focusing solely on the potential of the process of the CNNs yields the conclusion that DL, in general, has what it takes to discover significant features.

### 3.5. Grad-CAM Results

Selected images were processed from the trained FF-VGG19 network to inspect the decisive features’ location using the Grad-CAM algorithm. The training datasets and the selected images are presented in Table 13.

The Grad-CAM algorithm traces the convolutional layers and neurons participating strongly in formulating the final classification outcome. In this way, FF-VGG19 revealed the important areas of the original images, wherein the decisive features were discovered. Hence, the Grad-CAM algorithm helps interpret and evaluate the feature extraction process, which in many cases, has a valuable contribution to improving the model’s explainability.

For the SPNs depicted in Figure 7a, it can be noticed that the benign SPNs are highlighted in red, and their surroundings are not highlighted at all, in contrast with the malignant SPNs, where the red colourmap covers the entire image. It is demonstrated that the model sought information in the correct areas, although the discovered information could not be visualised, at least with the Grad-CAM algorithm.

It can also be noticed that the COVID-19 X-ray images were classified as such on the basis of discovered patterns in the high respiratory system. In contrast, normal cases were classified on the basis of features in other locations. This demonstrates that CNNs were looking for information in the right direction, as it has been proven that the novel coronavirus affects specific areas of the respiratory system. However, it can also be observed that some irrelevant features were extracted and misjudged as decisive ones.

The detection of COVID-19 in CT images has been questioned in the scientific community. For this reason, the highlighted areas of the CT images (Figure 7c) are not reliable. However, the CNNs also highlighted irrelevant areas (e.g., c1 outer left picture, c2 outer left picture). Therefore, it was assumed that the model was confused by the incomplete and irrelevant-to-COVID-19 information discovered in CT cases. Since the CT_COV dataset included few images, no further assumptions can be made at this stage.

For the myocardial perfusion polar maps, the CNNs were based on green areas of the initial image, which is the correct method. However, the initial image itself was not informative in a significant proportion of the subjects and led to false positives.

The highlighted areas in images depicting skin abnormalities (Figure 8) do not provide useful information regarding the important features. It was only confirmed that FF-VGG19 detected the area of interest correctly, as it was not pointing toward irrelevant locations. In combination with the activation maps of Figure 4, it is demonstrated that FF-VGG19 managed to exclude confusing patterns in the images, such as the surrounding hair.

## 4. Discussion

Training from scratch was superior to transfer learning in most of the utilised datasets and for the majority of the CNNs. Moreover, it was observed that FF-VGG19 effectively trained from scratch and transfer learning, providing a stable and robust framework for developing a general medical image classification model. The results indicate that the special features learned during the training from scratch are potential biomarkers, especially in cases where the amount of the training data allows for deep CNNs.

Next, the covariance shift methodology was followed to perform a deeper analysis of the extracted features. Specifically, it was investigated whether the learned features of a specific dataset (e.g., skin cancer) can be retained for predictions on a separate dataset of the same nature and kind. The results indicate that the DL model could extract important features from the source datasets and achieve top performance on alternative and unseen datasets. This yields strong evidence that CNNs can identify general and decisive image features of medical importance, which are met on a specific training dataset and refer to global features existing in images of the same domain. Activation maps were also visualised. The Grad-CAM algorithm was employed to shed light on those features. It was observed that the DL model extracted relevant and significant feature maps among the numerous extracted maps. Furthermore, the Grad-CAM algorithm revealed that the DL model correctly pointed to the valid areas of interest where human expertise focuses its attention when examining similar medical images.

The findings of the study agree with the findings of recent works. For example, Yang et al. [47] presented an attention-based explainability framework to detect important areas of interest in breast cancer histopathology images. Their framework, although not suggesting novel biomarkers, was able to identify the expected regions correctly. In PET imaging, Choi et al. [48] developed a deep learning-based cognitive signature of FDG brain PET adaptable for Parkinson’s disease (PD) and Alzheimer’s disease (AD). The two-dimensional projection mapping visualised the degree of cognitive dysfunction in agreement with the expected degree. In [49], the authors designed an explainable framework for automated skin lesion segmentation and classification. The visualised class activation maps verified that the model seeks important features in the correct locations of the image. Again, the visualisation could not suggest specific biomarkers but large regions. The review study of van der Velden et al. [50] discussed several recent studies that used explainable methods for various medical imaging modalities, including PET, MRI, and CT.

The contribution of this research is threefold. Firstly, a deep and extensive analysis of the extracted features of CNNs was conducted. This analysis yielded strong evidence that medical-specific features (i.e., image biomarkers) are extracted when training the proposed networks from scratch and observing large-scale datasets, especially in pathological findings. The outcome is derived from obvious features (e.g., colour, size, and shape). Moreover, the study performed a preliminary analysis of the feature maps and employed the Grad-CAM algorithm to inspect the suggested regions.

Secondly, the study identified the most suitable medical image data classification networks. More precisely, FF-VGG19 and Xception stood out in most classification tasks. On the other hand, DenseNet is ideal for skin malignancy detection.

Thirdly, this research proposed a novel modification of the traditional VGG19 network, which exploited the benefits of feature fusion via concatenation and outperformed every competitive network.

Lastly, we suggested a modification for further evaluation in larger-scale datasets of various medical applications.

### Limitations

One may highlight several limitations which fall beyond the scope of this study and raise general issues for the scientific community. Firstly, most of the available datasets are not large enough to successfully train deep and complex CNNs. Therefore, it is imperative that institutions and researchers collaborate to release well-labelled and large-scale datasets of various modalities. Secondly, while various datasets were utilised, the actual range of medical image modalities and diseases was not covered. Further research could be conducted to explore a wider range. Thirdly, since more state-of-the-art networks have emerged lately, a limitation of this study is that they were excluded.

Moreover, this study focused solely on automatic feature extraction methods. At the same time, several hybrid methodologies were recently proposed to both extract hand-crafted features of proven importance and let the CNNs automatically extract more features. Future studies could involve such methods to inspect the usefulness of the suggested features and how much they correlate with known biomarkers or other predefined image characteristics. For example, employing explainable ML algorithms, such as random forest and fuzzy cognitive maps, in integrated DL-ML frameworks could provide more details about the importance of the extracted features.

The field of explainable artificial intelligence was not entirely explored. For example, recent algorithms such as LIME and Grad-CAM++ were not evaluated. It is the authors’ intention to benchmark these methods in future studies.

## 5. Conclusions

In the present research study, DL models were assessed to inspect their capabilities in extracting and presenting potential image biomarkers. Firstly, it was observed that transfer learning via training from scratch is preferable over feature extraction via transfer learning when large-scale and good-quality data are present. Training from scratch makes it possible to detect special features which can be identified as image biomarkers. Secondly, the inspection of the extracted features utilising feature activation maps and algorithms revealing the location of those features verified that those methods are suitable, can improve the explainability of the models, and can provide an analysis of the extracted characteristics. Their results revealed that DL seeks potential biomarkers in the right direction. The deep layers of the networks indeed revealed those important features. Thirdly, the novel FF-VGG19 outperformed every state-of-the-art CNN when training from scratch, while it remained stable when utilised for transfer learning. Therefore, it could serve as a baseline model for many related medical image classification tasks, including PET scans, CT scans, X-ray scans, dermoscopy, and histopathological images.

## Figures and Tables

**Figure 1 diagnostics-12-02333-f001:**
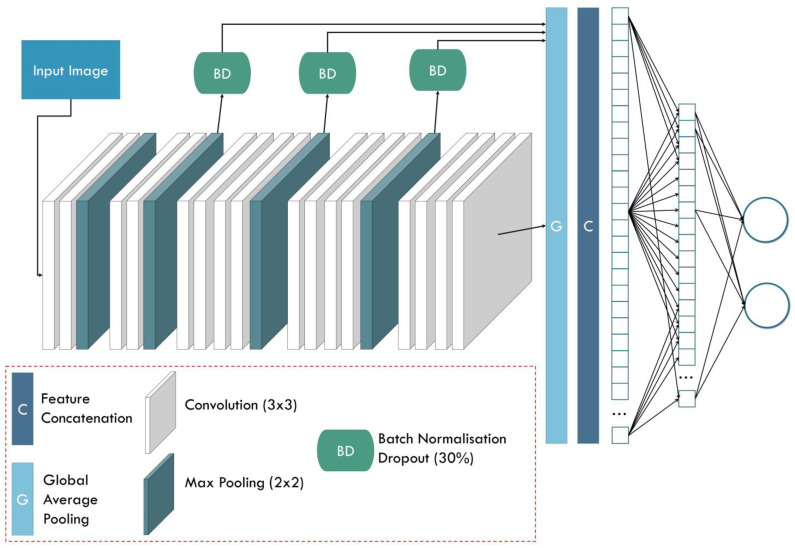
Structure of the FF-VGG19 network.

**Figure 2 diagnostics-12-02333-f002:**
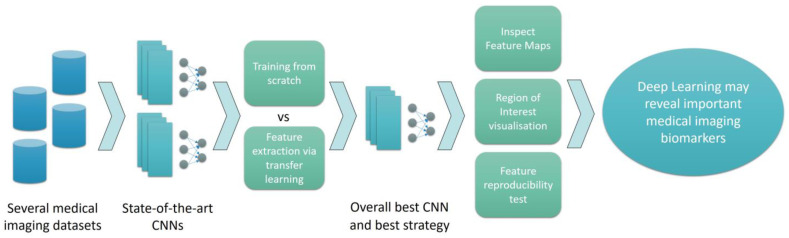
Methodology of the current research work.

**Figure 3 diagnostics-12-02333-f003:**
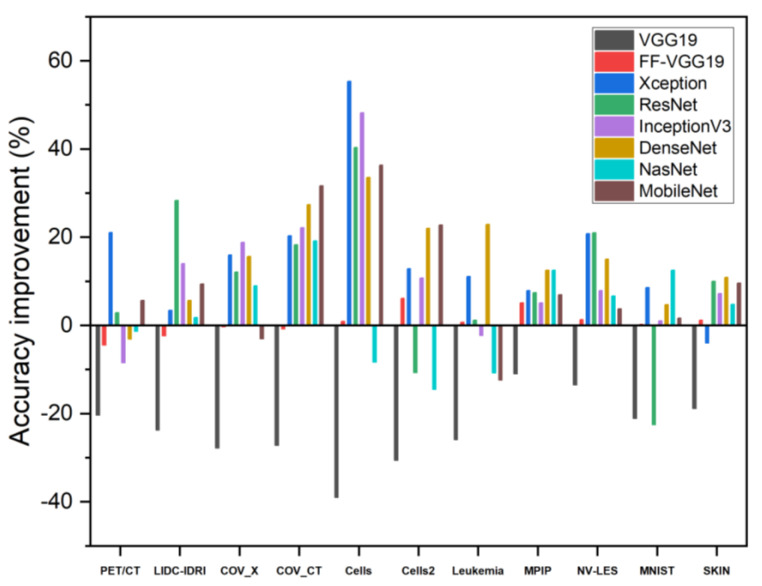
Accuracy improvement of the CNNs when training from scratch on the study’s datasets. It can be observed that, in most of the datasets, training from scratch was beneficial for the networks and improved their diagnostic capabilities.

**Figure 4 diagnostics-12-02333-f004:**
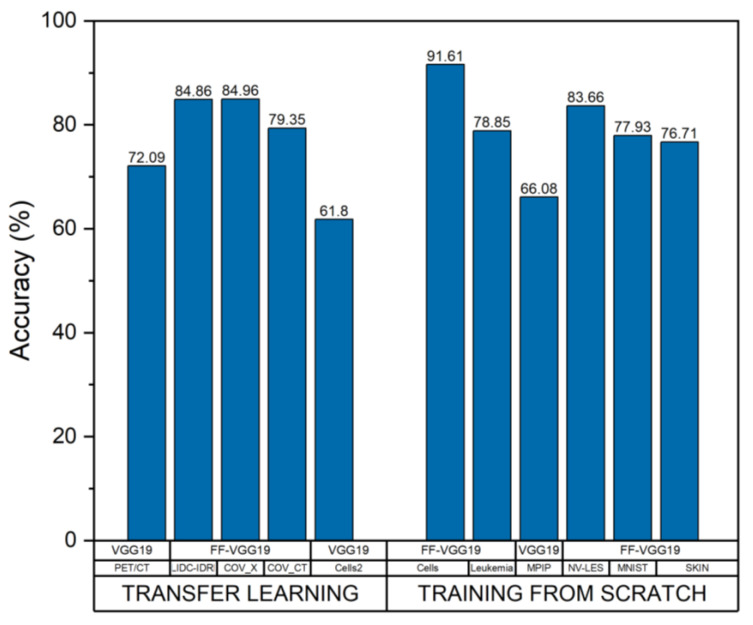
Best-performing CNNs and learning strategies for each of the experimental medical imaging datasets.

**Figure 5 diagnostics-12-02333-f005:**
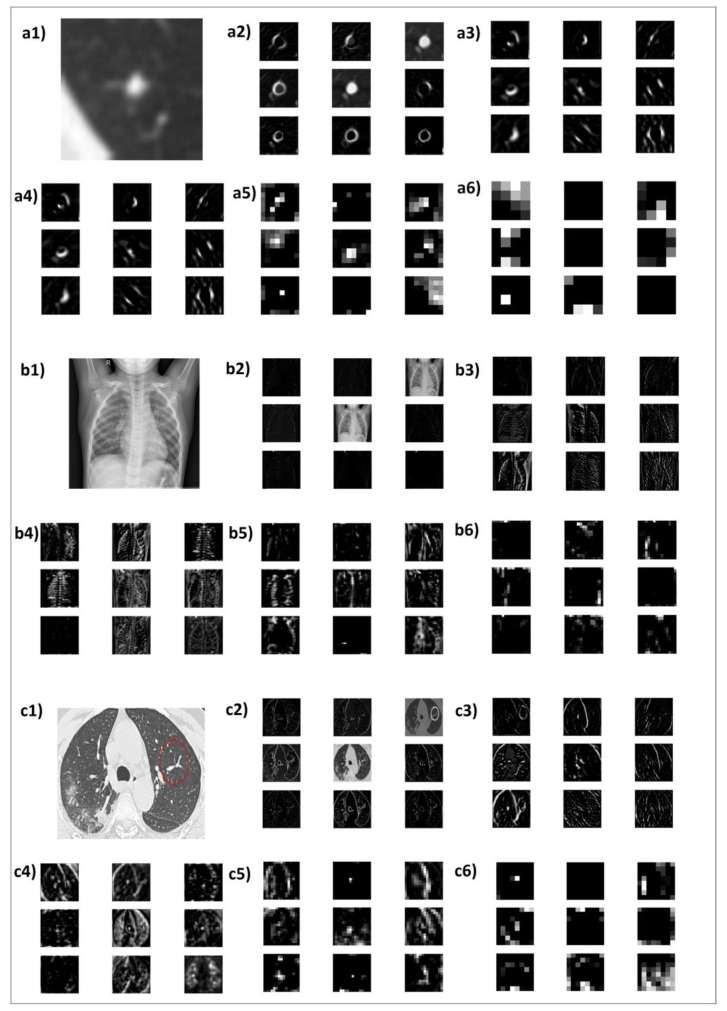
FF-VGG19 extracted feature maps of a random input image, where (**a1**–**a6**) refer to feature maps from the LIDC-IDRI + PET dataset, (**b1**–**b6**) refer to features maps from the COV_X dataset, and (**c1**–**c6**) refer to the COV_CT dataset.

**Figure 6 diagnostics-12-02333-f006:**
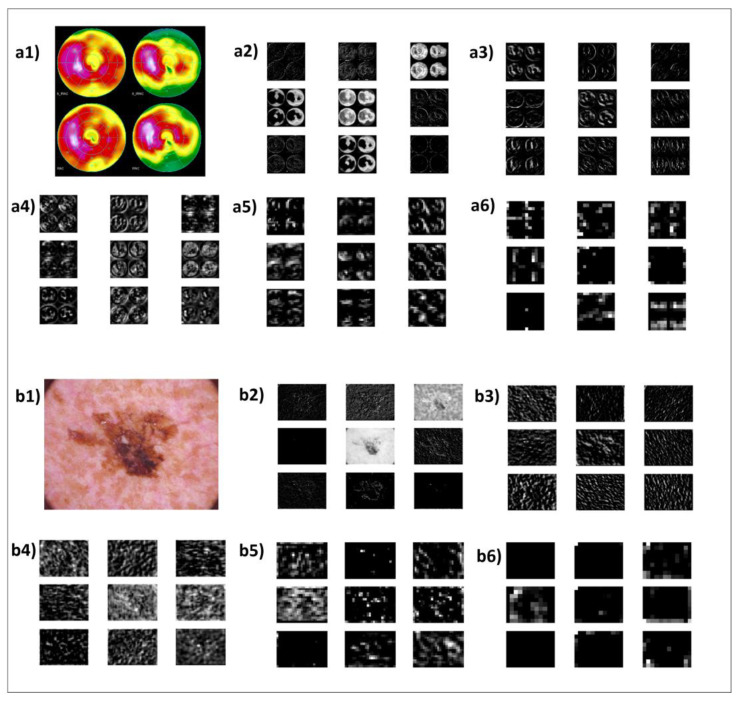
FF-VGG19 extracted feature maps of a random input image, where (**a1**–**a6**) refer to feature maps from the MPIP dataset, and (**b1**–**b6**) refer to features maps from the MNIST + SKIN + NVLES dataset.

**Figure 7 diagnostics-12-02333-f007:**
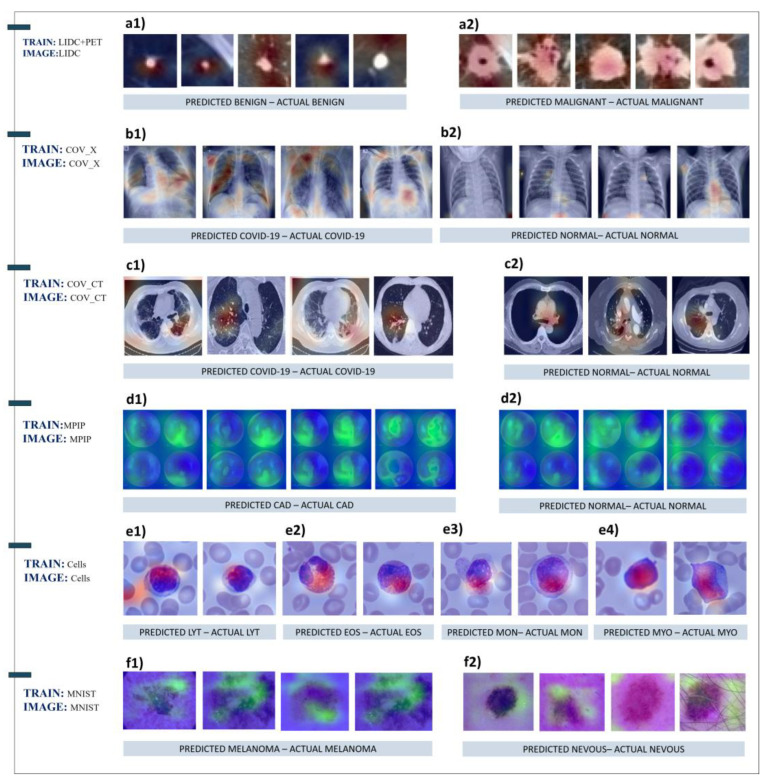
Grad-CAM results for the following test dataset samples: (**a**) LIDC, (**b**) COV_X, (**c**) COV_CT, (**d**) MPIP, (**e**) Cells, and (**f**) MNIST.

**Figure 8 diagnostics-12-02333-f008:**
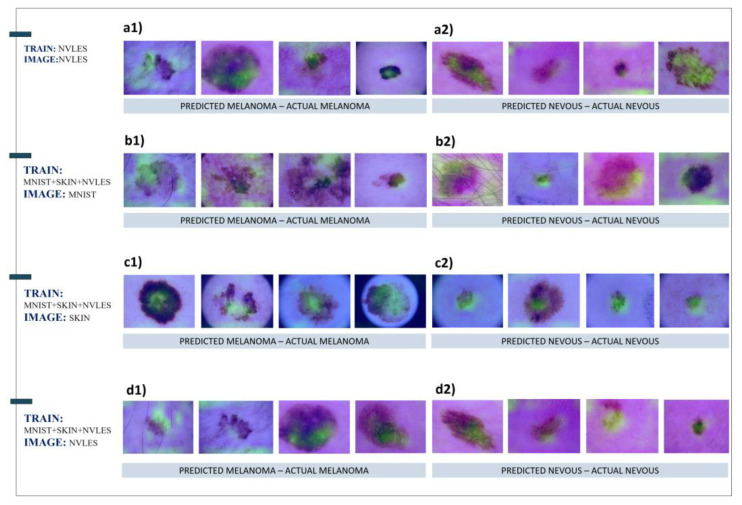
Grad-CAM results for the following test dataset samples: (**a**) NVLES, (**b**) MNIST, (**c**) SKIN, and (**d**) NVLES.

**Table 1 diagnostics-12-02333-t001:** Description of the study’s datasets.

Dataset Name	Modality and Domain	Image Type (Source)	Objective	System	Classes (Number of Images)
**PET/CT** [21]	PET/CT scans, including SPNs	Dicom	Distinguish between benign and malignant solitary pulmonary nodules	General Electric Healthcare: Discovery iQ3 sl16	Benign (61) Malignant (111)
**LIDC-IDRI** [29]	CT scans, including SPNs	Dicom	Distinguish between benign and malignant solitary pulmonary nodules	Various CT systems	Benign (620) Malignant (616)
**COV_X** [30]	X-ray scans of respiratory diseases	jpeg	Distinguish between COVID-19 and other diseases	Various X-ray systems	Pulmonary oedema (293) Pleural effusion (311) Obstructive pulmonary disease (315) Pulmonary fibrosis (280) COVID-19 (455) Bacterial and viral pneumonia (910) Normal cases (1341)
**COV_CT** [31]	CT scans of respiratory diseases	Dicom	Distinguish between COVID-19 and non-COVID-19 cases	Various CT systems	COVID-19 (349) Normal (397)
**Cells** [32]	Microscopic Images of cancerous cells	tiff	Distinguish between types	Precipoint GmbH, Freising, Germany	Eosinophil (424) Typical lymphocyte (3937) Monocyte (1789) Myeloblast (3268) Neutrophil (4419)
**Cells2** [33]	Microscopic Images of cancerous cells	bmp	Distinguish between types	Not available	Monocyte (20) Eosinophil (88) Lymphocyte (34)
**Leukaemia** [34]	Microscopic images of cancerous cells	jpeg	Haematological malignancy detection	Not available	Cancerous (8491) Normal (4037)
**MPIP** [35]	SPECT MPI polar maps of CAD	Dicom	Distinguish between healthy and coronary artery disease subjects	Infinia, Hawkey-4, GE Healthcare	CAD (136) Normal (80)
**NV-LES** [36]	Skin images for skin cancer recognition	jpeg	Distinguish between benign nevus and cancerous melanoma	Various	Benign (3000) Cancerous (3000)
**MNIST** [37]	Skin images for skin cancer recognition	jpeg	Distinguish among 5 skin abnormal classes	Various	Actinic keratoses (327) Basal cell carcinoma (514) Benign keratosis (1099) Melanoma (1113) Melanocytic nevi (6705)
**SKIN** [38]	Skin images for skin cancer recognition	jpeg	Distinguish among 3 skin abnormal classes	Various	Benign (1843) Melanoma (753)

**Table 2 diagnostics-12-02333-t002:** DL networks employed in the present study.

Network Name	Version	Year	Reference
Virtual Geometry Group (VGG)	VGG16	2015	[17]
Feature-Fusion VGG19	Baseline	2021	[35]
Residual Network	v.151	2015	[40]
Inception	v.3	2015	[41]
Xception	Baseline	2017	[42]
Densely Connected Network	v.121	2018	[43]
NasNet	Mobile version	2018	[44]
Mobile Network	v.2	2017	[45]

**Table 3 diagnostics-12-02333-t003:** Experiment setups for evaluating the training from scratch against feature extraction via transfer learning methodologies.

Sub-Experiment	Train Dataset	Test Dataset	Networks	Transfer Learning
1	PET/CT	PET/CT	VGG19, FF-VGG19 Xception, ResNet50, Inception V3, DenseNet 121, MobileNet v2, NasNet mobile	By training from scratch or by using “off-the-shelf” features
2	LIDC-IDRI	LIDC-IDRI		
3	COV_X	COV_X		
4	COV_CT	COV_CT		
5	Cells	Cells		
6	Cells2	Cells2		
7	Leukaemia	Leukaemia		
8	MPIP	MPIP		
9	NV-LES	NV-LES		
10	MNIST	MNIST		
11	SKIN	SKIN		

**Table 4 diagnostics-12-02333-t004:** Experiment setups for evaluating the models’ robustness to image device variation.

Experiment	Train Dataset	Test Dataset	Learning
1	LIDC-IDRI	PET/CT	Training from scratch
3	MNIST	NV-LES	Training from scratch
4	NV-LES	SKIN	Training from scratch
5	SKIN	MNIST	Training from scratch

**Table 5 diagnostics-12-02333-t005:** Classification accuracy (%) of employed CNNS, utilising off-the-shelf features. The table columns refer to the various networks employed for the classification. XCEPT refers to Xception, RES refers to ResNet, INCE refers to InceptionV3, DENSE refers to DenseNet, NAS refers to NasNet, and MOB refers to MobileNet. The evaluation is based on a 10-fold cross-validation.

Case	Train Dataset	Test Dataset	VGG19 (%)	FF-VGG19 (%)	XCEPT (%)	RES (%)	INCE (%)	DENSE (%)	NAS (%)	MOB (%)
1	PET/CT	PET/CT	72.09	55.19	48.72	47.12	55.28	62.71	55.28	51.79
2	LIDC-IDRI	LIDC-IDRI	80.97	84.86	78.57	50.16	60.10	71.91	50.72	64.97
3	COV_X	COV_X	79.92	84.96	44.63	51.57	58.56	60.10	52.90	56.03
4	COV_CT	COV_CT	78.16	79.35	53.89	52.01	48.79	39.94	44.10	40.34
5	Cells	Cells	86.70	90.74	34.56	30.88	24.53	39.49	28.59	30.38
6	Cells2	Cells2	61.80	41.14	46.80	40.52	38.57	24.90	41.47	25.28
7	Leukemia	Leukemia	72.47	77.92	67.77	67.73	64.48	46.03	61.17	63.35
8	MPIP	MPIP	63.24	61.01	51.85	49.53	45.37	45.83	46.75	47.68
9	NV-LES	NV-LES	78.46	82.36	56.98	49.25	56.48	61.03	49.00	53.46
10	MNIST	MNIST	74.59	77.74	65.71	68.71	62.90	68.65	54.89	68.72
11	SKIN	SKIN	71.38	75.57	65.89	65.55	64.75	63.11	61.01	62.94

**Table 6 diagnostics-12-02333-t006:** Classification metrics of the best CNN for each dataset when training with off-the-shelf features. Some metrics are not computed due to multiclass classification.

Case	Best-Performer	Accuracy (%)	Sensitivity (%)	Specificity (%)	AUC	F1-Score
1	VGG19	72.09	83.52	60.91	76.64	74.61
2	FF-VGG19	84.86	84.87	82.95	91.47	84.25
3	FF-VGG19	84.96	-	-	96.70	-
4	FF-VGG19	79.35	85.67	73.80	87.64	77.13
5	FF-VGG19	90.74	-	-	94.85	-
6	VGG19	61.80	-	-	63.58	-
7	FF-VGG19	77.92	78.25	60.29	81.23	77.27
8	VGG19	63.24	86.76	17.50	83.33	22.00
9	FF-VGG19	82.36	85.16	79.56	93.11	83.27
10	FF-VGG19	77.74	-	-	84.53	-
11	FF-VGG19	75.57	51.97	87.98	82.28	82.48

**Table 7 diagnostics-12-02333-t007:** Classification accuracy of the CNNs when training from scratch. The table columns refer to the various networks employed for the classification. XCEPT refers to Xception, RES refers to ResNet, INCE refers to InceptionV3, DENSE refers to DenseNet, NAS refers to NasNet, and MOB refers to MobileNet. The evaluation is based on 10-fold cross-validation.

Case	Train Dataset	Test Dataset	VGG19 (%)	FF-VGG19 (%)	XCEPT (%)	RES (%)	INCE (%)	DENSE (%)	NAS (%)	MOB (%)
1	PET/CT	PET/CT	51.76	50.72	69.77	50.00	46.77	59.58	53.92	57.45
2	LIDC-IDRI	LIDC-IDRI	57.21	82.52	81.96	78.45	74.10	77.58	52.50	74.35
3	COV_X	COV_X	52.11	84.64	60.61	63.63	77.36	75.72	61.86	52.95
4	COV_CT	COV_CT	50.93	78.57	74.16	70.24	70.91	67.29	63.27	71.98
5	Cells	Cells	47.68	91.61	89.85	71.20	72.74	73.05	20.24	66.68
6	Cells2	Cells2	31.19	47.23	59.66	29.80	49.28	46.90	26.95	48.00
7	Leukemia	Leukemia	46.57	78.58	78.85	68.86	62.15	68.9	50.37	50.91
8	MPIP	MPIP	52.27	66.08	59.72	56.94	50.46	58.33	59.25	54.62
9	NV-LES	NV-LES	64.96	83.66	77.75	70.24	64.38	76.05	55.66	57.21
10	MNIST	MNIST	53.52	77.93	74.28	46.20	63.91	73.34	67.40	70.38
11	SKIN	SKIN	52.52	76.71	61.92	75.56	71.95	73.99	65.77	72.52

**Table 8 diagnostics-12-02333-t008:** Classification metrics of the best CNN for each dataset when training from scratch.

Case	Best-Performer	Accuracy (%)	Sensitivity (%)	Specificity (%)	AUC	F1-Score
1	Xception	69.77	69.76	65.76	76.64	60.93
2	FF-VGG19	82.52	76.29	88.70	90.67	80.80
3	FF-VGG19	84.94	-	-	95.25	-
4	FF-VGG19	78.57	87.96	70.27	83.48	79.23
5	FF-VGG19	91.61	-	-	94.46	-
6	Xception	59.66	-	-	69.46	-
7	Xception	78.85	87.19	61.25	83.86	0.5
8	FF-VGG19	66.08	85.29	33.75	90.70	39.26
9	FF-VGG19	83.66	84.70	82.63	87.76	84.24
10	FF-VGG19	77.93	-	-	84.58	-
11	FF-VGG19	76.71	47.36	92.13	83.98	83.77

**Table 9 diagnostics-12-02333-t009:** FF-VGG19 and Xception network performance under medical image acquisition device variation.

Train Dataset	Test Dataset	FF-GG19 (%)	Xception (%)
LIDC-IDRI	PET/CT	74.41	70.93
MNIST	NVLES	69.93	63.92
SKIN	MNIST	56.25	85.25
NVLES	SKIN	61.64	66.86

**Table 10 diagnostics-12-02333-t010:** Analytical metrics for the Xception network when evaluating on alternate acquisition device.

Train Dataset	Test Dataset	ACC	SEN	SPE	F1	AUC
LIDC-IDRI	PET/CT	70.93	0.84	0.59	73.95	83.02
MNIST	NVLES	63.92	0.85	0.59	67.28	79.39
SKIN	MNIST	85.25	0.21	0.86	92.02	59.27
NVLES	SKIN	66.86	0.78	0.67	79.69	58.85

**Table 11 diagnostics-12-02333-t011:** Analytical metrics for the FF-VGG19 network when evaluating on alternate acquisition device.

Train Dataset	Test Dataset	ACC	SEN	SPE	F1	AUC
LIDC-IDRI	PET/CT	74.41	0.86	0.63	76.84	84.54
MNIST	NVLES	69.93	0.98	0.42	76.49	88.61
SKIN	MNIST	56.25	0.63	0.55	68.36	63.98
NVLES	SKIN	61.64	0.76	0.54	64.79	68.66

**Table 12 diagnostics-12-02333-t012:** The training dataset and the testing image of each visualised feature map group.

Training Dataset	Testing Images
LIDC-IDRI + PET	From LIDC-IDRI
COV_X	From COV_X
COV_CT	From COV_CT
Cells	From Cells
MPIP	From MPIP
MNIST	From MNIST
NV-LES	From NV-LES
MNIST + SKIN + NVLES	From NVLES
MNIST + SKIN + NVLES	From SKIN
MNIST + SKIN + NVLES	From MNIST
LIDC-IDRI + PET	From LIDC-IDRI
COV_X	From COV_X

**Table 13 diagnostics-12-02333-t013:** The training dataset and the testing image of each visualised Grad-CAM.

Training Dataset	Testing Images
LIDC-IDRI + PET	From LIDC-IDRI
COV_X	From COV_X
COV_CT	From COV_CT
Cells	From Cells
MPIP	From MPIP
MNIST	From MNIST
NV-LES	From NV-LES
SKIN	From SKIN
MNIST + SKIN + NVLES	From NVLES
MNIST + SKIN + NVLES	From SKIN
MNIST + SKIN + NVLES	From MNIST
LIDC-IDRI + PET	From LIDC-IDRI

## Data Availability

The PET/CT and MPIP datasets are not publicly available due to ethical reasons. The LIDC-IDRI, COV_X, COV_CT, Cells, Cells2, Leukaemia, NV-LES, MNIST, and SKIN datasets are publicly available.

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
