# Peer review of "Deep Learning Assessment for Mining Important Medical Image Features of Various Modalities"

_diagnostics, 2022, doi:10.3390/diagnostics12102333_

Round 1

Reviewer 1 Report

The study is an interesting one. Can the authors incorporate retianl fundus dataset? The authors evaluated the datasets with different methodologies like transfer learning, fine-tuning etc. Are these methodologies not explored earlier by other researchers? Is explaininiblity the only novelty in this regards? If not, the authors may compare their study with the SOTA studies in this domain in a table. Are the attention mechanism or ViT not relevant in this regards? Can a comparison be made with or without extracted features for the methodologies for the particular datasets?

Author Response

The study is an interesting one.

Response:

We thank the reviewer for acknowledging our work.

Can the authors incorporate retinal fundus dataset?

Response:

Unfortunately, the incorporation of the retinal fundus dataset is not a possibility, due to the lack of experience in such tasks. The author team is focused on PET/SPECT/CT, skin, and X-Ray imaging.

The authors evaluated the datasets with different methodologies like transfer learning, fine-tuning etc. Are these methodologies not explored earlier by other researchers?

Response:

Recent research reports acceptable results using these methodologies. For example, in SPN classification, both transfer learning and training from scratch yield exceptional sensitivity. It is the authors’ intention to re-evaluate these methodologies using the modified version of VGG19 (FF-VGG19), which is not the task of related works. Besides, testing these methods again has revealed some interesting conclusions. For example, we conclude that “Training from scratch or transfer learning is a suitable learning strategy in cases wherein a feature of vital medical importance can be the shape, the greyness level, or the size of a pathological finding” (Section 3.1).

Is explainability the only novelty in this regards? If not, the authors may compare their study with the SOTA studies in this domain in a table.

Response:

Comparisons with the state-of-the art CNNs are conducted across the study. For example, the study employs ResNet, Inception, Xception, and other networks for classifying the participating datasets. We have also compared our study with recent studies involving visual explanations across the manuscript. Please refer to the modified related works section and discussion. Since the intentions of the study is not to focus on the accuracy metrics, but to study the visual explanations, comparison with related works regarding the diagnostic accuracy would be redundant. Therefore, we preferred to discuss the key findings of the related works in separate paragraphs, rather than summarizing them to tables.

Are the attention mechanism or ViT not relevant in this regards? Can a comparison be made with or without extracted features for the methodologies for the particular data

Response:

Our models do not include the attention mechanism. We have included this limitation in the limitations section and we also presented the results of a related work that employs attention networks. Comparisons with the state-of-the art CNNs is conducted across the study. This includes comparing the results of training from scratch (learning new features) and transfer learning (using features from another domain).

Reviewer 2 Report

manuscript present feature extraction using DL methods. IT uses several secondary data and also implement various strategies and compares them. Simulation resullts are presented and results are promising. 

However i have following suggestion.

1. There is not clarity of pipeline followed or proposed. Diagram is missing

2. Generally the images will come incremetally in such problems once deployed in real time what about the modification to address the same. pls ref https://www.sciencedirect.com/science/article/abs/pii/S0957417418304731 for seeing how incremeal uses can be done

3. There is no proper validation method used.

4. what about XAI methods ohter than grad cam? ref to https://ieeexplore.ieee.org/abstract/document/9391727 for alternative method and select the best one

5. typos to be eliminated

6.  authors need to do more literature survey and write comparative study in literature sections. for eg many recent methods proposed for feature selection like hybrid, low cost etc ref https://link.springer.com/article/10.1007/s00170-022-09784-y and 

Author Response

  1. There is not clarity of pipeline followed or proposed. Diagram is missing

Response:

We have included an additional figure that describes the methodology of this study in more detail (Figure 1).

  1. Generally the images will come incremetally in such problems once deployed in real time what about the modification to address the same. pls ref https://www.sciencedirect.com/science/article/abs/pii/S0957417418304731 for seeing how incremeal uses can be done

Response:

Deployment of DL-based systems in real time is not a domain this paper focuses on. The present study intends to assess DL methods from a biomedical engineering perspective rather than a statistical and mathematical evaluation based solely on pre-defined accuracy metrics. The study evaluates the potential importance of medical imaging features extracted by several DL models. The reviewer is right to point out this perspective. However, the authors believe that it is not necessary to include discussion on this matter, because it may confuse the readers.

  1. There is no proper validation method used.

Response:

For all the experiments we used the ten-fold cross validation method for training and testing the proposed networks. This method guarantees that the networks are evaluated using unseen data. For the visual explanations, the author team analysed the results of the Grad-CAM algorithm, based on their experience in medical imaging (NDP > 10 years, NIP > 10 years, GSP>25 years). The authors argue that the validation method both in terms of the accuracy metrics and in terms of the visual explanations are adequate to reach the conclusions of this study.

  1. what about XAI methods ohter than grad cam? ref to https://ieeexplore.ieee.org/abstract/document/9391727 for alternative method and select the best one

Response:

It is true that other methods such as SAP, LIME, Grad-CAM++ are not evaluated here. This was out of scope of this study. Also, due to time constraints, adapting this study to utilise such algorithms is prohibitive and would require repeating all the experiments. Therefore, we have included this issue in the limitations section.

  1. typos to be eliminated

Response:

We have performed an analytical scan for typographical and grammatical errors and the language is improved.

  1. authors need to do more literature survey and write comparative study in literature sections. for eg many recent methods proposed for feature selection like hybrid, low cost etc ref https://link.springer.com/article/10.1007/s00170-022-09784-y and

Response:

We have improved our related work section. We have also included comparisons with key findings in recent literature. Since the evaluation is not based on accuracy metrics, such comparisons are held in terms of their visual findings. Please refer to the Discussion section.

Reviewer 3 Report

In this paper, author evaluated DL methods for their efficiency in revealing and suggesting potential image
biomarkers.  However, there are some limitations that must be addressed as follows.

1.        In both abstract and introduction Section, it is difficult to understand the novelty of the presented research work. These sections should be modified carefully. I suggest to divide section 1 into two section: introduction and related work. In addition, In Introduction section, the main contribution should be presented in the form of bullets.

2.        It is difficult to understand the methods of this work. Authors should include figures Methods Section. There should be framework in form of figure.

3.        Where is the details explanation of CNN, figure?

4.        The most recent work about medical image classification should be discussed as follows (‘A Two-Tier Framework Based on GoogLeNet and YOLOv3 Models for Tumor Detection in MRI’, and ‘A review on deep learning in medical image analysis’).

5.        More details should be included about data augmentation.

6.        It is better to include sentence before starting subsection 3.1.

7.        Figures are blurred and difficult to read. These should be improved.

8.        Captions of the Figures and tables not self-explanatory. These captions should be self-explanatory, and clearly explaining the figure. Extend the description of the mentioned figures and tables to make them self-explanatory.

9.        The whole manuscript should be thoroughly revised in order to improve its English.

10.     Future work should be included in conclusion with details explanation.

Author Response

  1. In both abstract and introduction Section, it is difficult to understand the novelty of the presented research work. These sections should be modified carefully. I suggest to divide section 1 into two section: introduction and related work. In addition, In Introduction section, the main contribution should be presented in the form of bullets.

Response:

We thank the reviewer for this suggestion. The Introduction is improved as per the reviewer’s comment. 

  1. It is difficult to understand the methods of this work. Authors should include figures Methods Section. There should be framework in form of figure.

Response:

We have included an additional figure that describes the methodology of this study in more detail (Figure 1).

  1. Where is the details explanation of CNN, figure?

Response:

We did not include figures representing the CNNs used in the study, because they are available publicly. State of the art CNNs were not modified. However, we did include a figure presenting the FF-VGG19 network, which is an innovative modification of VGG19 and is extensively evaluated in the present article.

  1. The most recent work about medical image classification should be discussed as follows (‘A Two-Tier Framework Based on GoogLeNet and YOLOv3 Models for Tumor Detection in MRI’, and ‘A review on deep learning in medical image analysis’)

Response:

Thanks for the suggestion. We have included the mentioned works in our manuscript, where we discuss the recent progress of DL in imaging tasks. Please refer to section 2.1

  1. More details should be included about data augmentation.

Response:

We have improved the data augmentation section in the revised version.

  1. It is better to include sentence before starting subsection 3.1.

Response:

We have included an introductory text in the revised version.

  1. Figures are blurred and difficult to read. These should be improved

Response:

We are sorry for this inconvenience. It may have happened due to the pdf compression. We will ensure the provision of higher resolution images in the revised manuscript submission.

  1. Captions of the Figures and tables not self-explanatory. These captions should be self-explanatory, and clearly explaining the figure. Extend the description of the mentioned figures and tables to make them self-explanatory.

Response:

We have modified the Table and Image captions to be more self-explanatory.

  1. The whole manuscript should be thoroughly revised in order to improve its English.

Response:

We have performed an analytical scan for typographical and grammatical errors and the language has been improved.

  1. Future work should be included in conclusion with details explanation.

Response:

We have discussed more about feature opportunities and directions in the limitations section.

Round 2

Reviewer 2 Report

suggested changes are not done by the authors.

Reviewer 3 Report

  The authors have addressed my all comments. I have no further comments. Therefore, this paper can be accepted in its present form.